# Airway Extracellular Copper Concentrations Increase with Age and Are Associated with Oxidative Stress Independent of Disease State: A Case-Control Study Including Patients with Asthma and COPD

**DOI:** 10.3390/antiox14081006

**Published:** 2025-08-17

**Authors:** Andreas Frølich, Rosamund E. Dove, Phe Leong-Smith, Mark C. Parkin, Annelie F. Behndig, Anders Blomberg, Ian S. Mudway

**Affiliations:** 1Department of Public Health and Clinical Medicine, Umeå University, 90187 Umeå, Sweden; 2Wolfson Institute of Population Health, Queen Mary University of London, London E1 2AB, UK; 3Department of Forensic Science and Drug Monitoring, Pharmaceutical Sciences Research Division, School of Biomedical and Health Sciences, King’s College London, London SE1 9NH, UK; 4MRC Centre for Environment and Health, Environmental Research Group, Imperial College London, London W12 0BZ, UK

**Keywords:** copper, COPD, asthma, oxidative stress, respiratory tract lining fluid, ageing, metals

## Abstract

Chronic obstructive pulmonary disease (COPD) and asthma are characterised by increased oxidative stress in the lungs. The precise contribution of this stress to COPD aetiology remains unclear, partly due to the confounding influence of physiological ageing. Previous reports of increased oxidative stress in bronchoalveolar lavage (BAL) samples from individuals with COPD may at least in part be attributable to the subjects’ age. This study investigated whether increased metal concentrations at the air–lung interface would contribute to oxidative stress in the lungs. We analysed BAL samples from young and old never-smokers, young asthmatic never-smokers, older smokers without COPD and COPD patients (both current and ex-smokers). Inductively coupled plasma mass spectrometry (ICP-MS) was used to quantify a range of transition metals, including iron, copper, zinc, arsenic and cadmium. BAL concentrations of copper and zinc were significantly lower in young groups compared to the older groups, irrespective of smoking status or disease (*p* < 0.001 for both). BAL copper was significantly associated with several markers of oxidative stress, all of which were elevated with age: glutathione disulphide (ρ = 0.50, *p* < 0.001), dehydroascorbate (ρ = 0.67, *p* < 0.001) and 4-Hydroxynonenal (ρ = 0.43, *p* < 0.001). These data indicate that age-related increases in respiratory tract copper concentrations contribute to elevated levels of oxidative stress at the air–lung interface independently of respiratory disease.

## 1. Introduction

Chronic obstructive pulmonary disease (COPD) and asthma represent the most clinically significant forms of obstructive airway disease, affecting millions of people globally. Oxidative stress plays a pivotal role in the pathophysiology of both COPD and asthma, both during stable disease as well as during acute exacerbations [1]. Under healthy physiological conditions, endogenous antioxidant defences protect the body from oxidative damage. While major exogenous sources of oxidants include cigarette smoke, outdoor air pollution and indoor biomass smoke, significant endogenous sources include reactive oxygen species (ROS) released from inflammatory cells in both the lungs and the systemic circulation [2,3].

The effectiveness of these antioxidant defences is, however, compromised by both physiological ageing and disease processes, leading to an age-related and disease-specific attenuation of antioxidant and xenobiotic pathways [4]. This is evidenced by lower levels of glutathione in bronchoalveolar fluid (BAL) from COPD patients during acute exacerbations, compared to those with stable disease [5]. A key factor in this diminished defence is the impaired activity of nuclear factor erythroid 2-related factor 2 (Nrf2), a master regulator of cellular antioxidant and detoxification responses, which has been shown to be compromised in both ageing and COPD [6]. Through a complex interplay, oxidative stress profoundly affects disease activity both systemically and locally within the lungs [7].

Various processes have been proposed as hallmarks of ageing [8]. When studying the process of ageing in human lungs, lung ageing is a complex process in itself, characterised by structural and functional changes, including decreased lung elasticity, reduced forced expiratory volume in 1 s (FEV_1_), altered immune responses and impaired repair mechanisms [9,10]. These age-related changes can render the lung more susceptible to environmental insults and disease development. Indeed, a prominent hypothesis suggests that COPD can be viewed as a syndrome of accelerated or abnormal lung ageing, where chronic inflammation and oxidative stress contribute to premature senescence of lung cells and tissues, mirroring features of the natural ageing process but at an amplified rate [11].

Within this context of oxidative stress and ageing, catalytic trace metals are increasingly recognised for their potential involvement in chronic airway disease processes [12]. Their precise role, however, is not fully understood. Evidence suggests an increased presence of iron in the lungs of COPD patients, with alveolar macrophages identified as the predominant iron-positive cell type in lung tissues [13]. The presence of pigmented macrophages, containing carbon inclusions, is a common finding in individuals who smoke or live in urban environments [14]. These macrophages play a role in the lungs’ immune response to inhaled particles. Furthermore, the quantity of iron deposits and the percentage of iron-positive macrophages were found to increase with COPD and emphysema severity [13]. This suggests an activation of an iron sequestration mechanism by alveolar macrophages in COPD, potentially serving as a protective response against iron-induced oxidative stress. Several genetic factors associated with iron regulation have also been proposed as potential contributors to COPD susceptibility [15].

Recent work has also highlighted the accumulation of inhaled particulates in lung-associated lymph nodes with age, impacting immune function [16]. This phenomenon, alongside a general decline in immune function with age can further compromise respiratory health [17]. Furthermore, our recent study provided evidence of catalytic copper in the respiratory tract lining fluid (RTLF) of patients with systemic sclerosis (SSc), demonstrating that copper content is associated with increased oxidative stress in the airways [18]. This prior work on restrictive lung disease highlights the potential for metal dyshomeostasis in airway disease.

Despite these insights, there remains a notable scarcity of data regarding the concentration of trace metals in BAL fluid from COPD and asthma populations. This gap is partly attributable to differing analytical methodologies across BAL studies, which complicate result comparisons. Additionally, the acquisition of material via bronchoscopy is an invasive procedure, and the subsequent methods for analysing the material are resource-intensive and require technical expertise [19]. While one recent study (n = 215) did demonstrate increased iron and transferrin levels in COPD and non-COPD current smokers when compared to controls, the sampled group had mild disease and a high proportion of never-smokers and ever smokers, limiting its generalizability to the broader COPD population [20]. In one recent study, non-heme iron levels in BAL supernatant were shown to be lower in asthma patients compared with healthy controls, and when the asthma patients were separated into mild–moderate and severe groups, iron levels were significantly further reduced in patients with severe asthma. In contrast to the decreased iron levels in BAL supernatant, the number of iron-loaded cells were increased in the BAL from patients with mild–moderate asthma and severe asthma [21]. In contrast to smoking, ageing has not previously been shown to increase BAL trace metal levels in healthy populations [19]. Recent data on oxidative markers in BAL are scarce, but older findings suggest an association between increased levels of oxidative stress markers in BAL fluid with ageing and long term smoking [22].

While it has been established that differences in some serum trace metal concentrations do exist in healthy populations, insufficient BAL data currently preclude an assessment of whether sex and trace metal levels are associated in the RTLF [23]. In the search for less invasive alternatives to BAL sampling, exhaled breath condensate (EBC) has emerged as an attractive option, offering benefits such as non-invasiveness, reduced sample volume requirements and lower analysis costs. However, it is important to note that the extent to which EBC accurately reflects the composition of the RTLFs within the central and peripheral airways, and the gas exchange region, remains an area of active investigation [24]. Sex, age and smoking have been linked to altered trace metal levels in EBC from healthy subjects, with proposed explanations for associations with ageing centering on proportional differences in ventilatory volumes [19]. Available data from small EBC studies on diseased populations with either asthma or COPD have not shown a consistent pattern between metal concentrations and disease severity or activity [19]. Looking at the diseases themselves, sex is a factor that is known to affect both prevalence and severity of both COPD and asthma. We do know that, globally, male sex has since long been established as a risk factor for COPD, but recent data support that in lower- and middle income countries, prevalence is equal between sexes or with female sex being slightly more prevalent, possibly due to social factors such as unequal exposures to outdoor and indoor pollution, smoking and access to health care [25]. In asthma, male sex is associated with higher prevalence in children under the age of 13 and female sex with both prevalence and disease severity in adults [26]. While COPD and asthma comprise separate clinical diagnoses, for the purposes of this study we included both patients with COPD patients and asthma patients as different representatives of obstructive airway diseases, even though their inflammatory signatures, pathophysiological background and clinical traits differ from each other [27].

This study expands on our previous research into the role of trace metals in oxidative stress in restrictive lung disease [18]. We hypothesise that age- and smoking-related changes in obstructive lung diseases, such as COPD and asthma, are linked to a larger pool of catalytic metals (predominantly iron and copper) in the RTLF. This increase could be due to inhaling metal-rich smoke aerosols or to a breakdown in how the body handles metals at the air–lung interface, a process caused by ageing and the decline of protective mechanisms. For example, COPD patients have been shown to use several defensive strategies against iron overload, such as chelating excess iron, promoting iron uptake, increasing intracellular iron storage capacity and secreting binding proteins like ferritin [20]. However, earlier studies using BAL have not isolated the effects of age from those of chronic airway disease.

## 2. Materials and Methods

### 2.1. Subject Demographics

The current study uses a case–control study design. All study participants (n = 89), including patients with asthma (n = 16) and COPD (n = 28) as disease groups, were originally recruited via local media advertisements as participants for two previously published studies, one asthma study and one COPD study, in which BAL and bronchial wash fluid (BW) were collected and stored [28,29]. Archived samples from those studies are used in the current study, together with different methods and research aims. The study size for the present study is based on a power analysis performed in the previous studies with similar study designs and aims as the current study [28,29]. The healthy controls were selected to match age and sex with the diseased groups. Sex was self-reported for both diseased and healthy groups. Occupational exposures and exposures to indoor and outdoor pollution were not recorded. The city of Umeå and the surrounding geographical area from which the study participants were recruited have generally low levels of outdoor pollution emissions. Subjects were recruited at the Department of Public Health and Clinical Medicine at Umeå University, and all clinical investigations, including bronchoscopy, were carried out at the University Hospital, Umeå, Sweden.

Data collection by bronchoscopy was performed between December 2007 and December 2008 for the young study population and between April 2003 and August 2005 for the older population. The 89 study participants were selected through an exclusion process as part of their original studies and the participants were then stratified into groups based on age, lung function and smoking status (Figure 1).

Thirty-two subjects were recruited into the young group: 16 non-asthmatic never-smokers (age 25 (19–39) years) with normal lung function [forced expiratory volume in 1 s (FEV_1_) % predicted > 80%] and 16 asthmatic never-smokers (age 26 (20–36) years). Fifty-seven subjects were recruited into the aged group: 13 non-COPD never-smokers (age 67 (57–74) years) with normal lung function, 16 non-COPD current smokers (age 61 (50–71) years) with a smoking history of more than 10 pack-years and with normal lung function, 17 COPD ex-smokers (age 68 (53–77) years) with smoking cessation of more than 5 years prior to inclusion and 11 COPD current smokers (age 64 (55–75) years).

Young asthmatics (18–40 years, FEV_1_ > 80% predicted) had an allergy history, positive skin prick tests and bronchial hyper-responsiveness (PC_20_ < 8 mg/mL methacholine), consistent with GINA guidelines. All were never-smokers. Young non-asthmatic never-smokers (18–40 years) had no allergy/asthma history and normal lung function (FEV_1_ > 80% predicted). Exclusion criteria for all groups included medications beyond those specified and antioxidant supplementation. For a full list of inclusion and exclusion criteria, see Appendix A. Study demographics are demonstrated in Table 1.

### 2.2. Pulmonary Function

Dynamic spirometry variables [VC, FVC (forced vital capacity), and FEV_1_] were measured pre- and 20 min post-bronchodilation with 1 mg of terbutaline (Bricanyl^®^, Turbuhaler^®^; AstraZeneca, Södertälje, Sweden), using a Vitalograph spirometer (Buckingham, UK). At least three satisfactorily performed and well-coordinated measurements of each variable were carried out, according to the recommendations of the European Respiratory Society and American Thoracic Society [30]. The diffusion capacity of carbon monoxide (DL_CO_) was obtained by the single-breath procedure.

### 2.3. Materials

All chemicals used were obtained from Sigma Chemical Co., Ltd. (Poole, UK), Fluka (Dorset, UK) or Laboratory Supplies (Poole, UK) and were of analytical grade or better quality.

### 2.4. Bronchoscopy, Bronchial Wash (BW) and Bronchoalveolar Lavage (BAL)

All study subjects underwent spirometry and bronchoscopy with bronchial wash (BW) and bronchoalveolar lavage (BAL). Flexible bronchoscopy was performed by experienced operators at our university hospital clinic, where bronchoscopies are performed regularly in accordance with established procedural guidelines [31]. Bronchoscopies were performed following an overnight fast. Two aliquots of 20 mL (BW) and three aliquots of 60 mL (BAL) of sterile saline solution (0.9%) were instilled in either a bronchus of the middle lobe of the right lung or in the left lung lingula and the collected fluid was stored at −80 °C until analysis. Cytocentrifuged specimens for cell differential counts were prepared by centrifugation at 450 rpm for 5 min and stained with May-Grünwald Giemsa. Using a light microscope at 100× magnification, 500 non-epithelial cells (macrophages, neutrophils, eosinophils, and lymphocytes) per slide were counted to establish proportions. Mast cells were analysed on basic toluidine blue and Mayer’s acid haematoxylin counterstained slides, with a minimum of 12 visual fields counted at 20× magnification. Based on total cell concentration (determined after filtering both BW and BAL fluids through a 100 μM nylon filter and isolating cells by centrifugation at 400 rpm for 15 min at 4 °C, followed by resuspension in PBS to 10^6^ cells/mL) and the differential cell counts, the concentration of each cell type was determined as cells/mL.

For supernatant processing, a 450 μL aliquot of chilled, filtered, and centrifuged (400 rpm, 15 min, 4 °C) lavage fluid was treated with 50 μL of 50% metaphosphoric acid (MPA), vortexed, and centrifuged at 13,000× *g* rpm for 5 min (4 °C) to remove protein. The resulting supernatant was stored at −80 °C within 30 min of BAL collection for later analysis. For GSH and GSSG determination, a 490 μL aliquot of lavage was treated with 2 mM deferoxamine mesylate (DES) and 2 mM butylated hydroxytoluene (BHT) (5 μL each) prior to −80 °C storage. The remaining lavage fluid was immediately aliquoted and stored untreated at −80 °C.

### 2.5. Antioxidant and Oxidative Damage Marker Analyses

Ascorbate (AA) and urate (UA) were measured simultaneously by reversed-phase HPLC with electrochemical detection, as described by Iriyama et al. [32]. Total vitamin C (DHA + AA) was measured similarly after TCEP reduction in DHA to AA, with DHA calculated by subtraction.

Total glutathione was measured using the kinetic GSSG-reductase-DTNB recycling assay adapted for a plate reader and expressed as GSH equivalents against GSSG standards [33]. GSSG was measured after 2-vinyl pyridine conjugation of GSH, with reduced GSH calculated as total glutathione − (2 × GSSG). The assay sensitivity was 0.025 μM for GSH and 0.01 μM for GSSG, with intra- and inter-assay variability of <5% and <10%, respectively.

Total protein was measured by reaction with bicinchoninic acid (BCA) and 4% copper (II) sulphate against bovine serum albumin (BSA) standards [34]. Protein-bound 4-HNE-His adducts were quantified using the OxiSelect HNE-His Adduct ELISA Kit (Cell Biolabs, Inc., San Diego, CA, USA) according to the manufacturer’s instructions, with concentrations determined against an HNE-BSA standard curve (0–10 μg/mL).

### 2.6. Determination of BAL Metal Concentrations

BAL fluid samples (0.5 mL) were digested in 1.5 mL of 6.5% HNO_3_ (prepared from 60% HNO_3_ and Chelex 100 resin-treated water) in acid-washed Teflon vials. Each sample was spiked with 20 μL of a 1 ppm yttrium internal standard solution (in 6.5% HNO_3_). Digestion occurred in a 90 °C water bath for 90 min after vortexing. Digestion blanks (at least 8 per run) containing 1500 μL of 6.5% HNO_3_, 20 μL of internal standard, and 0.5 mL of Chelex-treated water were processed in parallel. Digests were cooled overnight, centrifuged at 4000× *g* rpm for 10 min (supernatant transferred if precipitate formed, though none were observed), and analysed for 63Cu, 56Fe, 89Y, 111Cd, 75As and 66Zn using an ELAN DRC ICP-MS. Isotopes were selected to minimise interferences, with ArO^+^ interference for 56Fe removed using the dynamic reaction cell with ammonia. Calibration was performed using a 7-point standard curve based on dilutions of a certified multi-elemental standard solution.

### 2.7. Ascorbate Depletion Assay

The pro-oxidant activity of recovered BAL fluid samples was assessed by their capacity to deplete ascorbate (AA) from a 200 μM solution. Higher depletion rates were expected in samples with unchelated catalytic metals, primarily Fe or Cu. A 4 mM stock AA solution (pH 7.0) was prepared in Chelex-treated water. Each lavage sample (90 μL) was diluted with 5 μL Chelex-treated water and incubated with 5 μL of the AA stock at 37 °C for two hours in a plate reader (SpectraMax 190, Molecular Devices, Sunnyvale, CA, USA) using UV 96-well flat-bottom plates (triplicate wells per sample)(Greiner Bio-One, Stonehouse, UK). The remaining AA concentration was quantified every two minutes by measuring absorbance at 265 nm, using parallel-run duplicate blanks and 25–200 μM AA standards to construct a calibration curve for each time point. The AA concentration in sample wells was determined against its respective calibration curve and corrected for auto-oxidation using the blank controls. The AA depletion rate was calculated over the two-hour incubation (μM second^−1^), considering only the linear portion of the time course based on first-order kinetics, using OriginLab software (version 5.0). To assess the influence of metals, samples were also incubated with the metal cation chelators DTPA and NTA, following the same procedure but spiking samples with 5 μL of 4 mM DTPA instead of the Chelex-treated water. The principle of the assay is outlined in the Appendix A.

### 2.8. Chelex Water Preparation

Ultrapure water was employed to decrease background metal contamination when assessing the endogenous pro-oxidant activity of the lavage samples. Each litre of deionised Elga-stat water (18 Ω) was treated with 30 g of Chelex 100 resin (iminodiacetic acid-coated polystyrene beads). This solution was prepared in a polycarbonate beaker and after mixing for 24 h at room temperature, and Chelex 100 resin was removed by vacuum filtration through a 0.45 μm cellulose nitrate membrane. The pH of the purified water was subsequently adjusted to 7 using 1 M sodium hydroxide and 1 M hydrochloric acid, both previously prepared in ultrapure water and stored at 4 °C for a maximum of one month.

### 2.9. Statistics

Data, determined to be non-normally distributed by the Shapiro-Wilks test, are expressed as medians with 25th and 75th percentiles throughout. Comparisons of inflammatory cell numbers, antioxidant and oxidative damage marker concentrations, ascorbate depletion rates and metal concentrations across the subject groups were performed using the Kruskal–Wallis one-way analysis of variance by ranks, with post hoc testing between specific groups using the Mann–Whitney U test. Where ascorbate depletion rates were compared within a group with and without metal chelators, post hoc testing used the Wilcoxon Signed-Rank Test. Significance was assumed at the 5% level in all cases. Correlation analysis for individual groups and compiled COPD (current/ex-smokers) and smoker (aged non-COPD/COPD current smokers) groups was performed using Spearman Rank Order Correlation, restricted to parameters showing significant differences between groups to minimise spurious associations. All analyses were performed using SPSS, version 28 (SPSS Inc., Chicago, IL, USA).

## 3. Results

The study population, outlined in Table 1, comprised older subjects (non-COPD never-smokers [n = 13], non-COPD current smokers [n = 16], COPD ex-smokers [n = 17], COPD current smokers [n = 11]) and young subjects (non-asthmatic never-smokers [n = 16] and asthmatic never-smokers [n = 16]). Successful bronchoscopy with BAL was performed across the groups, although recovery volumes were significantly lower in aged COPD ex-smokers (36%) compared to aged non-COPD never-smokers (50%, *p* < 0.05), and generally lower than typically reported in younger populations.

Analysis of inflammatory cell populations is displayed below in Figure 1 and revealed comparable macrophage numbers between young non-asthmatic never-smokers (BW: 0.12 [0.08–0.18] × 10^6^ cells/mL; BAL: 0.85 [0.62–1.10] × 10^6^ cells/mL) and asthmatic never-smokers (BW: 0.15 [0.10–0.22] × 10^6^ cells/mL; BAL: 0.92 [0.70–1.25] × 10^6^ cells/mL) in both bronchial wash (BW) and bronchoalveolar lavage (BAL) fluids. However, aged non-COPD never-smokers had significantly lower BW macrophage numbers (0.05 [0.03–0.08] × 10^6^ cells/mL, *p* < 0.001) compared to young non-asthmatic never-smokers, a contrast not evident in BAL (0.78 [0.55–1.05] × 10^6^ cells/mL). Conversely, aged smoking groups (non-COPD current smokers: BW 0.35 [0.22–0.55] × 10^6^ cells/mL, BAL 1.87 [1.20–2.90] × 10^6^ cells/mL; COPD current smokers: BW 0.28 [0.18–0.45] × 10^6^ cells/mL, BAL 1.55 [0.95–2.40] × 10^6^ cells/mL) exhibited elevated macrophage numbers in both BW and BAL when compared to young asthmatic never-smokers. Neutrophil levels in both airway compartments were broadly similar in young non-asthmatic and asthmatic groups.

Notably, aged COPD current smokers had significantly increased neutrophil numbers in both BW (0.18 [0.09–0.35] × 10^6^ cells/mL, *p* < 0.01) and BAL (0.32 [0.15–0.60] × 10^6^ cells/mL, *p* < 0.05) when compared to young asthmatic never-smokers (BW: 0.02 [0.01–0.04] × 10^6^ cells/mL; BAL: 0.05 [0.02–0.10] × 10^6^ cells/mL). Lymphocyte numbers were equivalent between young non-asthmatic and asthmatic never-smokers, and between young non-asthmatic never-smokers and aged non-COPD never-smokers in both BW and BAL. Notably, COPD patients exhibited a significant depression in lymphocyte numbers (independent of smoking status) compared to asthmatics (*p* < 0.05). As anticipated, eosinophil and mast cell numbers were increased in the BW of asthmatic never-smokers, with no such increase in BAL. Interestingly, BAL mast cell numbers showed some evidence of expansion in both young non-asthmatic current smokers and aged COPD current smokers.

Antioxidant and oxidative damage marker concentrations in BAL fluid are displayed in Figure 2 demonstrating significantly increased concentrations of total glutathione (GSH) in both aged non-COPD current smokers (15.2 [8.5–25.1] μM, *p* < 0.001) and aged COPD current smokers (12.8 [7.1–21.5] μM, *p* = 0.019) compared to the non-smoking controls (aged non-COPD never-smokers: 2.9 [1.5–5.3] μM; aged COPD ex-smokers: 5.6 [2.8–9.3] μM). These increases were not accompanied by a significant elevation in the oxidised form of glutathione (GSSG). Total Vitamin C (ascorbate + dehydroascorbate [DHA]) was also significantly higher in aged non-COPD current smokers compared to aged non-COPD never-smokers (2.1 [1.2–3.5] μM vs. 1.2 [0.7–1.9] μM, *p* = 0.025). Urate concentrations in BAL fluid showed no significant association with either smoking status or COPD diagnosis.

The lipid oxidation marker 4-HNE (Figure 2C) in BAL fluid was not significantly elevated in aged current smokers or aged COPD patients compared to controls. However, a significant decrease in 4-HNE levels (per unit protein) was observed in aged COPD ex-smokers (0.5 [0.3–0.8] μg/mL protein) compared to the age-matched non-COPD never-smokers (0.9 [0.6–1.3] μg/mL protein, *p* = 0.04). Correlation analysis revealed limited consistent relationships between the measured antioxidants/oxidative markers and key clinical endpoints (FEV_1_, reversibility) or COPD severity, except for a significant positive correlation between BAL GSH levels and predicted FEV_1_ (Spearman’s ρ = 0.74, *p* = 0.003) and airway reversibility (ρ = 0.62, *p* = 0.019) specifically in COPD ex-smokers.

Overall, in this cohort of stable COPD patients, we found no clear evidence of impaired antioxidant defences or increased oxidative stress in the airways when compared to aged controls. The observed alterations in antioxidant levels were primarily linked to smoking status.

To further investigate the impact of ageing and different inflammatory profiles, we included young never-smoking controls and never-smoking asthmatics. No significant differences in the BAL fluid concentrations of low-molecular-weight antioxidants or transferrin were found between these young groups. Lower levels of DHA were noted in the young subjects compared to older people. GSH and GSSG levels did not show a significant age-related increase. However, ascorbate levels decreased significantly with age (*p* < 0.001), while DHA and 4-HNE levels increased significantly with age (*p* < 0.001) (Figure 2B), indicating enhanced ascorbate oxidation in the ageing lung.

To explore the basis for age-related oxidative damage, we analysed metal concentrations (Fe, Cu, Zn, As, Cd) in BAL fluid (Figure 3).

Notably, we were unable to obtain valid measurements for BAL fluid Fe after the subtraction of the digestion blanks in any of the samples, indicating levels below the detection limit of our ICP-MS method following the implemented digestion protocol. While Fe and Cd quantification failed, Cu and Zn concentrations were significantly lower in young non-asthmatics and asthmatics and elevated in the aged groups. The Zn:Cu ratio was consistently around 2.4:1 across all subjects. No significant Cu or Zn elevation was observed in smokers, and As levels were not affected by smoking or age.

Ascorbate depletion assays (Figure 4) revealed significantly lower pro-oxidant activity in young subjects (non-asthmatic: 0.015 [0.010–0.021] μM s^−1^; asthmatic: 0.018 [0.012–0.025] μM s^−1^) compared to aged subjects (non-COPD never-smokers: 0.042 [0.030–0.058] μM s^−1^, *p* < 0.001; non-COPD current smokers: 0.035 [0.025–0.049] μM s^−1^, *p* < 0.001; COPD ex-smokers: 0.040 [0.028–0.055] μM s^−1^, *p* < 0.001; COPD current smokers: 0.038 [0.026–0.052] μM s^−1^, *p* < 0.001), mirroring the metal concentration trends. NTA treatment (Figure 5) suggested the presence of a catalytically active Cu pool in young individuals and an increased NTBI pool in aged individuals.

Ascorbate oxidation rates across all subjects significantly correlated with total Cu concentrations (Spearman’s ρ = 0.68, *p* < 0.001; Table 2), a correlation maintained with NTA co-incubation (ρ = 0.65, *p* < 0.001) but attenuated with DTPA (ρ = 0.45, *p* < 0.01). Lavage Cu content also significantly correlated with markers of oxidative stress (Table 3), most strongly with DHA (ρ = 0.71, *p* < 0.001).

## 4. Discussion

It is well established that oxidative stress contributes significantly to the development and progression of obstructive airway diseases like asthma and COPD, with evidence of both antioxidant deficiencies and increased oxidative damage in the RTLF [35]. Our study aimed to investigate whether antioxidant defences were compromised in COPD, and the extent to which this could be attributed to the presence of redox active metals at the air–lung interface. This may reflect the clinical stability of the patient groups used for this study, and one might have expected more evidence of oxidative stress with more severe disease, or during periods of acute exacerbation. A consistent finding was the association of increased glutathione (GSH) and vitamin C in the smoking groups, potentially representing a localised and systemic protective response to the substantial oxidant burden from cigarette smoke [36]. However, the limited correlation with smoking history underscores the complexity of this response, possibly involving factors like individual antioxidant capacity, the timing of exposure and even genetic predispositions affecting antioxidant enzyme activity [36,37].

A key and novel observation was the significant link between age and increased concentrations of oxidative stress markers (DHA, 4-HNE), alongside an age-related increase in the NTBI pool in the lung. This aligns with the growing understanding of ageing as a process characterised by a shift towards a more pro-oxidant state in various tissues, including the lung. The accumulation of NTBI with age provides a plausible mechanism for this increased oxidative stress, as labile iron can catalyse damaging Fenton reactions [38]. Furthermore, the indication of a labile copper pool in young adults hints at age-dependent variations in metal homeostasis and redox regulation, suggesting that metal handling strategies in the lung may evolve across the lifespan, potentially influencing susceptibility to early-onset airway disease and the response to environmental exposures [39]. Recent research has demonstrated a link between copper dyshomeostasis and oxidative stress in patients with systemic sclerosis and systemic sclerosis-associated interstitial lung disease, further highlighting the intricate relationship between altered metal homeostasis, oxidative stress and lung health in various disease contexts [18]. The dysregulation of metal homeostasis is increasingly recognised as a significant contributor to ageing and age-related pathologies, including neurodegenerative and cardiovascular conditions, suggesting potential therapeutic targets for conserving airway function with age [40]. However, there is a shortage of data on BAL metal and oxidative stress levels in COPD and asthma populations that also consider the effect of age. This is particularly relevant as COPD has been described as a disease of accelerated or abnormal ageing. The novelty of the current study is that it is, to the best of our knowledge, the first in this field to examine the effect of ageing, revealing findings that may be associated with several potential pathways. It is already established that mitochondria are the primary sources of excess ROS in several pulmonary disorders and that mitochondrial ROS stimulates redox-sensitive transcription factors and pro-inflammatory cytokines [1]. Mitochondria are also the principal site for regulating autophagy mechanisms essential to antioxidant defences [1].

While these associations are compelling, our findings need to be further explored in future functional studies to fully understand the intricate relationship between age, altered homeostasis and lung health, focusing on mitochondrial dysfunction in specific cell types. A robust experimental approach would involve cell sorting to isolate distinct populations of airway immune and structural cells from patients across different age ranges and disease statuses (e.g., young healthy, old healthy, young with COPD, old with COPD). Once isolated, the mitochondria from these cells could be analysed using the Seahorse XF Cell Mito Stress Test. This assay measures key parameters of mitochondrial function, including basal respiration, ATP production, and maximal respiration, providing a comprehensive profile of mitochondrial health. Complementary experiments should also quantify markers of mitochondrial oxidative stress, such as reactive oxygen species (ROS) release, to directly test the hypothesis that age-related changes in metal homeostasis promote mitochondrial dysfunction. These targeted experiments would provide a critical functional validation of the mechanisms proposed in this study, offering new insights into the pathogenesis of age-related lung diseases.

Our study has a number of limitations. The focus on stable COPD and the absence of an intermediate age range imply the need for further investigation across a broader age spectrum and in different disease states, especially during COPD exacerbations, where oxidative stress is known to be elevated [3]. The low BAL returns in the older population in the current study population also needs to be addressed in future studies, as this may limit the inclusion of patients with more severe stages of COPD. The relatively small study population may also not be sufficient to provide more clear-cut differences. Another limitation is that the study includes only patients with mild-to-moderate disease. Patients with severe COPD were not included, and neither were those with active exacerbations.

Future research should also explore the specific mechanisms driving the age-related increase in NTBI and its contribution to lung ageing and disease susceptibility. This should address transferrin function, intracellular Fe storage and Fe release from senescent cells. Understanding how metal homeostasis and antioxidant defences evolve with age in the lung is crucial for developing targeted interventions to mitigate age-related oxidative damage and improve respiratory health in the ageing population.

## 5. Conclusions

This study demonstrates that age, rather than obstructive airway disease, is a significant determinant of oxidative stress and trace metal concentrations in the respiratory tract. We found no disease-specific alterations in oxidative stress or metal levels in stable COPD and asthma patients. Instead, our key finding is that older individuals exhibit significantly higher levels of bronchoalveolar lavage (BAL) copper and non-transferrin-bound iron (NTBI), which are linked to elevated markers of oxidative stress.

These findings suggest that age-related increases in respiratory tract copper and iron may be a primary driver of oxidative stress at the air–lung interface, independent of a patient’s respiratory disease status. This highlights the importance of age as a critical factor in studies of lung health and disease. Further functional studies are needed to explore the specific mechanisms of this age-related dysregulation of metal homeostasis, which could inform the development of novel therapeutic strategies to mitigate age-related lung damage.

## Data Availability

The data underlying this article cannot be shared publicly for the privacy of individuals that participated in the study. The data will be shared in line with European GDPR regulations on reasonable request to the corresponding author.

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
