# Peer review of "Airway Extracellular Copper Concentrations Increase with Age and Are Associated with Oxidative Stress Independent of Disease State: A Case-Control Study Including Patients with Asthma and COPD"

_antioxidants, 2025, doi:10.3390/antiox14081006_

Round 1
Reviewer 1 Report
This study investigates the correlation between oxidative stress, metal homeostasis, and COPD/asthma, with a particular focus on BAL changes and their association with age-related alterations. While the study does not require major revisions, the authors could strengthen the mechanistic understanding by discussing potential pathways such as age-related mitochondrial ROS (mitoROS) production, the role of autophagy, or other cellular mechanisms involved in redox imbalance. Additionally, incorporating or suggesting future functional studies, such as assays for ROS production or cell-based models. This would help clarify the biological relevance of metal accumulation and its impact on lung health.
No detailed comments, can be accepted in present form
Author Response
Comment 1: This study investigates the correlation between oxidative stress, metal homeostasis, and COPD/asthma, with a particular focus on BAL changes and their association with age-related alterations. While the study does not require major revisions, the authors could strengthen the mechanistic understanding by discussing potential pathways such as age-related mitochondrial ROS (mitoROS) production, the role of autophagy, or other cellular mechanisms involved in redox imbalance. Additionally, incorporating or suggesting future functional studies, such as assays for ROS production or cell-based models. This would help clarify the biological relevance of metal accumulation and its impact on lung health.
Response 1: On behalf of the team of authors I would like to thank you for your expert opinion. Your valuable feedback will improve the manuscript further. Your comments are very useful.
We have tried to strengthen the mechanistic understanding by including references to mitochondrial ROS and autophagy, as well as suggesting future functional studies (page nr 19, paragraph 1, line 5 – until completion of paragraph 2). Hopefully this highlights the relevance of metal accumulation and its impact on human health better.
Reviewer 2 Report
Comments and suggestions
- In the present study, building upon our understanding of the role of trace metals in oxidative stress and the impact of ageing on lung defence mechanisms, and extending prior work on restrictive lung disease such as SSc27 we hypothesise that smoking and age related changes in obstructive lung diseases like COPD and asthma are linked to an increased catalytic metal pool (predominantly iron and copper) within the RTLF. This increase could arise either from the inhalation of metal-rich smoke aerosols or from a dysregulation of metal handling at the air-lung interface caused by the ageing process and the associated decline in protective pathway. Which pathway is associated with this ageing process should be described in the introduction.
- Introduction, some references are very old, which should be updated with recent references relevant to your study.
- Inclusion and exclusion criteria should be presented in a table for better understanding.
- The current study uses a case-control study design. All study participants (n=89), which includes patients with asthma (n=16) and COPD (n=28) as diseased groups, were originally recruited via local media advertisements as participants for two previously published studies, in which BAL and bronchial wash fluid (BW) were collected and stored. Why choose this way asthma (n=16) and COPD (n=28)? What is the rationale?
- All material and chemical names should be checked carefully (Manufacturer, City, State, etc..?
- Please check Table 1: the heading font appears to be overlapping. Also, ensure the headings are in bold as necessary.
- Please ensure that all figures are clearly presented and comply with the journal’s formatting requirements
- The novelty of the study should be more clearly emphasized, highlighting how it differs from previous research.
- Conclusion: make concise and avoid repetition. Check if it's necessary to write this. We did observe significantly lower BAL copper concentrations in young groups compared to the aged groups, irrespective of smoking or disease. BAL copper was significantly associated with several markers of oxidative stress: glutathione disulphide (ρ =0.50, P<0.001), dehydroascorbate (ρ =0.67, P<0.001) and 4-Hydroxynonenal (ρ =0.43, P<0.001), indicating that age-related increases in respiratory tract copper concentrations contribute to elevated levels of oxidative stress at the air-lung interface independently of respiratory disease. Please check and revise.
- Typos and grammatical errors throughout the manuscript should be checked carefully.
- Please check the similarity that overlapping with some published papers.
Comments and suggestions
- In the present study, building upon our understanding of the role of trace metals in oxidative stress and the impact of ageing on lung defence mechanisms, and extending prior work on restrictive lung disease such as SSc27 we hypothesise that smoking and age related changes in obstructive lung diseases like COPD and asthma are linked to an increased catalytic metal pool (predominantly iron and copper) within the RTLF. This increase could arise either from the inhalation of metal-rich smoke aerosols or from a dysregulation of metal handling at the air-lung interface caused by the ageing process and the associated decline in protective pathway. Which pathway is associated with this ageing process should be described in the introduction.
- Introduction, some references are very old, which should be updated with recent references relevant to your study.
- Inclusion and exclusion criteria should be presented in a table for better understanding.
- The current study uses a case-control study design. All study participants (n=89), which includes patients with asthma (n=16) and COPD (n=28) as diseased groups, were originally recruited via local media advertisements as participants for two previously published studies, in which BAL and bronchial wash fluid (BW) were collected and stored. Why choose this way asthma (n=16) and COPD (n=28)? What is the rationale?
- All material and chemical names should be checked carefully (Manufacturer, City, State, etc..?
- Please check Table 1: the heading font appears to be overlapping. Also, ensure the headings are in bold as necessary.
- Please ensure that all figures are clearly presented and comply with the journal’s formatting requirements
- The novelty of the study should be more clearly emphasized, highlighting how it differs from previous research.
- Conclusion: make concise and avoid repetition. Check if it's necessary to write this. We did observe significantly lower BAL copper concentrations in young groups compared to the aged groups, irrespective of smoking or disease. BAL copper was significantly associated with several markers of oxidative stress: glutathione disulphide (ρ =0.50, P<0.001), dehydroascorbate (ρ =0.67, P<0.001) and 4-Hydroxynonenal (ρ =0.43, P<0.001), indicating that age-related increases in respiratory tract copper concentrations contribute to elevated levels of oxidative stress at the air-lung interface independently of respiratory disease. Please check and revise.
- Typos and grammatical errors throughout the manuscript should be checked carefully.
- Please check the similarity that overlapping with some published papers.
Author Response
On behalf of the team of authors I would like to thank you for reviewing our work, giving your time and your expert opinion. We have used your valuable feedback to improve our manuscript further.
Please find the detailed responses below and the corresponding revisions/corrections highlighted/in track changes in the re-submitted files.
Comment 1: In the present study, building upon our understanding of the role of trace metals in oxidative stress and the impact of ageing on lung defence mechanisms, and extending prior work on restrictive lung disease such as SSc27 we hypothesise that smoking and age related changes in obstructive lung diseases like COPD and asthma are linked to an increased catalytic metal pool (predominantly iron and copper) within the RTLF. This increase could arise either from the inhalation of metal-rich smoke aerosols or from a dysregulation of metal handling at the air-lung interface caused by the ageing process and the associated decline in protective pathway. Which pathway is associated with this ageing process should be described in the introduction.
Response: We agree with this comment and we have added information (page 3, paragraph 3, line ”In contrast to smoking…” until end of paragraph) and (page 4, paragraph 2, line ”This increase…” until end of paragraph). There is a number of pathways that have been demonstrated in COPD populations but have not isolated the effect of age specifically (page 4, lines "For example, COPD patients have been shown to use several defensive strategies against iron overload, such as chelating excess iron, promoting iron uptake, increasing intracellular iron storage capacity and secreting binding proteins like ferritin[20]. However, earlier studies using BAL have not isolated the effects of age from those of chronic airway disease"). In the discussion, following comments from both reviewers, we have added reference to mitochondrial ROS and autophagy as well as suggestions for future studies, as the ageing-specific pathways need clarification (page nr 19, paragraph 1, line 5 – until completion of paragraph 2).
Comment 2: Introduction, some references are very old, which should be updated with recent references relevant to your study.
Response: We agree and we have updated the citation list, removing a number of older citations and added several recent ones. The older citations that we decided to keep were judged to be essential to the current manuscript, even if they are not recent.
Comment 3: Inclusion and exclusion criteria should be presented in a table for better understanding.
Response: We agree and we have included inclusion and exclusion criteria in the supplement.
Comment 4: The current study uses a case-control study design. All study participants (n=89), which includes patients with asthma (n=16) and COPD (n=28) as diseased groups, were originally recruited via local media advertisements as participants for two previously published studies, in which BAL and bronchial wash fluid (BW) were collected and stored. Why choose this way asthma (n=16) and COPD (n=28)? What is the rationale?
Response: We agree and we have added information (page 4, paragraph 1, line ”While COPD and asthma…” until end of paragraph). These subjects were originally included in previous studies, and were part of well-defined study cohorts. During the study design we chose to have two "representations" of obstructive lung diseases to include as much of the clinical panorama of this disease group, hopefully increasing the study impact further.
Comment 5: All material and chemical names should be checked carefully (Manufacturer, City, State, etc..?
Response: We have checked all material and chemical names and they seem to be correct. We made some minor adjustments to spelling (California changed to US, page 7, last paragraph under 2.5. Antioxidant and Oxidative Damage Marker Analyses).
Comment 6: Please check Table 1: the heading font appears to be overlapping. Also, ensure the headings are in bold as necessary.
Response: We agree that the font appears to be overlapping. We have tried several different ways to represent the heading fonts. We want to make it as clear as possible and there is little space because the six groups are lined next to eachother. We also wanted the numenclature to be consistent with the the tables and figures, where the information is spelled out in full, like ”Young non-asthmatic never-smokers”. We have reduced the font size to hopefully improve clarity and added bold in headings. We agree that this is not optimal but we see problems with the other alternatives as well and we think this is the most acceptable.
Comment 7: Please ensure that all figures are clearly presented and comply with the journal’s formatting requirements.
Response: We have tried our best to meet the formatting requirements. We have removed S3 and S4 as they were not mentioned in the manuscript and they were blurry. The remaining figures are clearly presented, complying with the formatting requirements.
Comment 8: The novelty of the study should be more clearly emphasized, highlighting how it differs from previous research.
Response: We have added to the discussion, hopefully adding emphasis on the novelty and added value to this area of research (Page 18, bottom paragraph, from line ”However, there is a shortage of data …” to page 19, first paragraph, line ”… potential pathways”) as well as (page 3, paragraph 3, line ”In contrast to smoking, ageing has not previously been shown to increase BAL trace metal levels in healthy populations”). Thank you for your expert opinion.
Comment 9: Conclusion: make concise and avoid repetition. Check if it's necessary to write this. We did observe significantly lower BAL copper concentrations in young groups compared to the aged groups, irrespective of smoking or disease. BAL copper was significantly associated with several markers of oxidative stress: glutathione disulphide (ρ =0.50, P<0.001), dehydroascorbate (ρ =0.67, P<0.001) and 4-Hydroxynonenal (ρ =0.43, P<0.001), indicating that age-related increases in respiratory tract copper concentrations contribute to elevated levels of oxidative stress at the air-lung interface independently of respiratory disease. Please check and revise.
Response: We fully agree. This is repetitive and unnecessary. We have revised the conclusion section, hopefully improving the clarity of the text (multiple edits; see revised manuscript). Thank you for pointing this out.
Comment 10: Typos and grammatical errors throughout the manuscript should be checked carefully.
Response: We have corrected a number of grammatical errors and typos. We have made multiple edits to the text that was repetitive or not clearly expressed. Thank you for pointing this out.
Comment 11: Please check the similarity that overlapping with some published papers.
Response: We have checked and have to our knowledge not found significant overlaps.
Response to Comments on the Quality of English Language: We have made multiple edits to the text that was repetitive or not clearly expressed. We believe that now the English is improved and expresses the research more clearly. If parts of the text needs further language improvement it would be useful if these could be specified.
Round 2
Reviewer 2 Report
Thank you for your efforts in improving the manuscript.
Thank you for your efforts in improving the manuscript.